# Pediatric versus Adult Medulloblastoma: Towards a Definition That Goes beyond Age

**DOI:** 10.3390/cancers13246313

**Published:** 2021-12-16

**Authors:** Joseph R. Wooley, Marta Penas-Prado

**Affiliations:** Neuro-Oncology Branch, National Cancer Institute, National Institutes of Health, Bethesda, MD 20892, USA; joseph.wooley@nih.gov

**Keywords:** medulloblastoma, clinical trials, age, molecular diagnostics

## Abstract

**Simple Summary:**

Medulloblastoma is a rare brain tumor that affects children and adults. Treatment with surgery, radiation, and chemotherapy currently cures most patients; however, ~30% of all patients have poor clinical outcomes despite treatment. Prospective clinical trials have historically excluded older patients, while recent advances in molecular diagnostics have enhanced our understanding of tumorigenesis. The aim of this literature review is to discuss the history of clinical trials in medulloblastoma and to argue in favor of prioritizing molecular drivers of disease as trial inclusion features rather than an arbitrary age cutoff.

**Abstract:**

Medulloblastoma is a rare malignant brain tumor that predominantly affects children but also occurs in adults. The incidence declines significantly after age 15, and distinct tumor molecular features are seen across the age spectrum. Standard of care treatment consists of maximal safe surgical resection followed by adjuvant radiation and/or chemotherapy. Adjuvant treatment decisions are based on individual patient risk factors and have been informed by decades of prospective clinical trials. These trials have historically relied on arbitrary age cutoffs for inclusion (age 16, 18, or 21, for example), while trials that include adult patients or stratify patients by molecular features of disease have been rare. The aim of this literature review is to review the history of clinical trials in medulloblastoma, with an emphasis on selection criteria, and argue in favor of rational and inclusive trials based on molecular features of disease as opposed to chronological age. We performed a scoping literature review for medulloblastoma and clinical trials and include a summary of those results. We also discuss some of the significant advances made in understanding the molecular biology of medulloblastoma within the past decade, most notably the identification of four distinct subgroups based on gene expression profiling. We will also cite the recent experiences of childhood leukemia and the emergence of tissue-agnostic therapies as examples of successes of rationally designed, inclusive trials translating to improved clinical outcomes for patients across the age spectrum. Despite the prior trial history and recent molecular advances outcomes remain poor for ~30% of medulloblastoma patients. We believe that defining patients by the specific molecular alterations their tumors harbor is the best way to ensure they can access potentially efficacious therapies on clinical trials.

## 1. Background

The term medulloblastoma was first used in 1925 by Drs. Cushing and Bailey, who reviewed a series of ~400 cases of gliomas and identified 29 cerebellar tumors occurring mainly in children (ages 2–28, median 8 years) [1]. Their series included five patients who were 18 years or older. There was uncertainty about the origin of these tumors, and they were noted to express different histological markers than gliomas or neuroblastomas. They were coined “medulloblastomas” due to their central, intracerebellar location, from the Latin *medullo* for “marrow” or “central part”.

Further research led to a reclassification system proposed by Dr. Rorke in 1983 that grouped medulloblastomas with other histologically similar primary CNS neoplasms composed of undifferentiated neuroepithelial cells and occurring primarily in infancy and childhood [2]. Collectively, these lesions were given the name “primitive neuroectodermal tumors” (PNET).

The World Health Organization began publishing consensus guidelines for the classification of CNS tumors in 1979, and these have evolved as our understanding of medulloblastomas has grown [3]. Prior to the 2000 update, evidence was accumulating that medulloblastomas and PNETs differed in a variety of ways (including sites of origin and treatment responses). Medulloblastomas were re-classified as a distinct subset of embryonal tumors and four different histological subtypes were recognized (classic, desmoplastic, extensive nodularity, and large cell). The classic subtype consists of small round cells with round/ovoid nuclei and represents >70% of cases, the desmoplastic/nodular and extensive nodularity subtypes contain varying levels of nodular neurocytic differentiation and reticulin deposition, and the anaplastic/large cell subtypes is characterized by large discohesive cells with prominent nucleoli, cytologic pleomorphism, and frequent mitoses [4,5]. The emergence of large-scale tumor DNA and RNA sequencing data in the late 2000s led to the realization that medulloblastomas compose multiple distinct subgroups with unique molecular genetic features [4]. Many groups proposed slightly different schemas, and these were refined into four consensus groups in 2012: SHH-activated, WNT-activated, Group 3, and Group 4 (Figure 1) [6].

The 2016 WHO update took the novel step of combining both “histologically defined” and “genetically defined” tumors to formulate integrated diagnoses [4]. Medulloblastomas were classified into one of five distinct molecular subgroups (the fifth group results from the SHH-activated tumors being split into TP53-wild type and TP53-mutant) and one of four histological subgroups: classic, desmoplastic/nodular, anaplastic/large cell, or medulloblastoma with extensive nodularity. The 2021 edition of the WHO Guidelines will again combine “molecularly defined” and “histologically defined” tumor features, defining four molecular subgroups (WNT-activated, SHH-activated and TP53-wildtype, SHH-activated and TP53-mutant, and non-WNT/non-SHH) and retaining the same four histological groups, while now considering them as “patterns” [7]. Additionally, the guidelines now also recognize distinct subtypes within the four principal molecular groups based on methylation and transcriptome profiling. SHH-activated tumors are now grouped into four different subtypes (alpha, beta, gamma, and delta) with distinct age profiles and outcomes. For example: SHH-beta tumors have a median age of onset around two years and are associated with a poor prognosis, while SHH-delta tumors have a median age of onset of 26 years and a better prognosis (Figure 2) [8,9]. Importantly, most tumor subtypes are seen across the entire age spectrum (from infants to older adults), even if at a lower frequency in certain age groups (Figure 2).

Although there is variation in incidence amongst different age cohorts, it is important to note that almost all tumor molecular subgroups and subtypes are present in all age cohorts. The data for the age at diagnosis is contained in the Supplemental Data section of Cavalli (2017) [9].

## 2. Materials and Methods

This scoping critical review was conducted via PubMed, Embase, and Web of Science by identifying articles regarding medulloblastoma diagnosis, treatment, and outcomes. Searches for original and review literature were conducted by J.W. from August until November of 2021 and included titles and abstracts published in English from 1925 through August of 2021. General search terms (including both Medical Subject Headings (MeSH) and free text words) included “medulloblastoma”, “adult medulloblastoma”, “medulloblastoma subgroups”, and “medulloblastoma clinical trials”. Bibliographic references of selected articles were also reviewed and referenced according to their relevance. Emphasis was given to high-profile works, particularly reports of clinical trials, as well as WHO guidelines, practice guidelines, and consensus statements.

## 3. Epidemiology

Based on an incidence of 1.58 (1.50–1.67) per million individuals (Surveillance, Epidemiology and End-Results database, SEER) and a total US population of around 330 million there are ~500 new cases of medulloblastoma diagnosed in the US each year [10]. The peak age of incidence is ~5 years and children are affected at a rate about ten times that of adults, with an incidence of 5.96 per million amongst ages 1–9 and 0.58 per million amongst adults 19 and over. These incidences have been stable since at least the 1970s.

These estimates are in line with data from the Central Brain Tumor Registry of the United States (CBTRUS), which estimated that from 2013 to 2017 there were an annual average of 293 new cases among children aged 0–14, 126 cases amongst adolescents and young adults aged 15–39, and 31 cases in adults aged 40 and over [11]. Although rare overall and often out of the public eye (especially in adults), these numbers are broadly on the scale with other rare cancers (such as malignant ovarian teratomas, ~500/cases/year in US) [12] and well-known genetic disorders such as Spinal Muscular Atrophy (~400 births per year) and Cystic Fibrosis (~1000 new diagnoses per year) [13,14].

The incidence of specific tumor subgroups varies based on age. Overall, Group 4 tumors are the most common (~35–40%), followed by SHH-activated (~30%), Group 3 (~20–25%), and WNT-activated (~10%) [15]. However, amongst adults these percentages vary due to an increase in the presence of SHH and a paucity of Group 3 variants, such that in adults approximately 60% of tumors are SHH-activated, followed by 25% Group 4, and 15% WNT-activated [16,17,18].

## 4. Treatment and Outcomes

Accurate diagnosis and staging includes MRI of the brain and spine (including contrasted, diffusion-weighted, and apparent diffusion coefficient (ADC) calculated images) cytology of the cerebral spinal fluid to detect for possible leptomeningeal spread, and immunohistochemical and advanced molecular analyses of tumor specimens [19,20]. Medulloblastomas are staged by the Chang system [21], which accounts for degree of tumor infiltration and the presence (and extent) of metastases, with poor prognoses associated with higher degrees of local tumor involvement in adults and the presence of metastatic disease in children [22,23]. Tumors in adults are more likely to be located laterally within the cerebellum, presumably due to the increased proportion of SHH tumors [24,25].

Current standard of care treatment for newly diagnosed medulloblastoma consists of maximal safe surgical resection followed by post-operative craniospinal radiation (CSI) and systemic chemotherapy. The largest differences in treatment between adult and pediatric patients arise out of differential toxicities. Radiation therapy is damaging to the developing nervous system, and is withheld in patients under 18 months, while cytotoxic chemotherapy is generally tolerated better by younger patients, with adults experiencing more adverse events and requiring delays and treatment cessation more frequently, particularly after craniospinal radiation treatment [26,27]. Historically, given the toxicities associated with cycles of multi-drug cytotoxic chemotherapy regimens and the lack of randomized data showing benefit in adult patients, adults with non-disseminated disease were typically recommended treatment with surgery followed by CSI alone. The change in practice to incorporate chemotherapy in all adults has not happened until recently, as we will discuss later; however, management remains heterogeneous and not all adults are treated with chemotherapy at initial diagnosis, with high variability among institutions and providers [28]. Although the rationale behind not using chemotherapy was sound, the best available evidence suggests a benefit, indicating that our assumptions can lead to suboptimal treatment in the absence of clinical trial data.

Radiation therapy consists of CSI plus a local boost, with variations based on age and risk status. High risk pediatric patients typically receive 36 Gray (Gy) of CSI followed by a posterior fossa boost, while average risk patients receive 23.4 Gy of CSI followed by a boost restricted to the tumor bed [28,29,30]. Adults typically receive 36 Gy of CSI and a boost to either the posterior fossa or the tumor bed [19].

Systemic chemotherapy is given in multiple cycles of combinations of cytotoxic drugs, either post-operatively and prior to CSI (as in the Taylor regimen, discussed later) or in combination with and following CSI, as is the Packer regimen (also discussed later) [31,32]. At present both children and adults are recommended to receive systemic chemotherapy, although the regimens and doses vary based on risk stratification and individual patient factors [19].

Outcomes with current standard of care treatment are generally good, with five-year overall survival (OS) rates over 70% across the age spectrum [33,34]. However, this figure captures marked heterogeneity, and the identification and validation of prognostic and predictive biomarkers is an area of active investigation.

Prognosis varies by subgroup, with WNT-activated tumors associated with the best prognosis (long-term survival rates over 90% in children) and Group 3 tumors generally having the worst prognosis, with long-term survival rates near 50%. SHH-activated and Group 4 tumors are both intermediate, with long-term survival rates around 75% [35]. Amongst adults poor prognostic factors include large cell/anaplastic histology [18] and Group 4 tumors, which were found associated with 5 year progression-free survival (PFS) of ~45% (vs. ~65% for WNT-activated, 62% for SHH-activated, and 80% for Group 3) [16,17]. Similarly, WNT-activated tumors do not appear to have such a favorable prognosis as in children [36]. Amongst SHH-activated tumors mutations in TP53, MYC-N amplifications, PTCH1 mutations, and multiple chromosome abnormalities (3p loss, 10q loss, and 17p loss) are associated with poor prognosis [16,37,38,39]. Furthermore, distinct molecular markers can be seen across both patient age and disease subgroups, such as MYC-N amplifications in all non-WNT activated tumors or OTX2 amplifications, common in both Group 3 and 4 tumors [40].

Treatment has improved markedly over the last 50 years, driven largely by collaborative clinical trials. However, most of these trials are conducted in pediatric populations only. Clinicaltrials.gov was accessed on 29 October 2021, and queried for “medulloblastoma” and trials that were in “recruiting” status; 60 trials resulted and were systematically evaluated. Seven were non-therapeutic (registry or imaging) studies. Of the remaining 53 therapeutic studies, only 13 (18.9%) are enrolling older adults (over 30 years of age), and key aspects are summarized in Table 1.

## 5. Clinical Trials

The basic standard of care, surgery followed by CSI, remained mostly unchanged for many decades after it was first proposed by Drs. Cushing and Bailey and lead to five-year survival rates of ~50% at best as of the 1960s [41].

The first reports of systemic chemotherapy in the 1960s reported patients with recurrent medulloblastoma who responded to vincristine [42,43]. In a 1972 review, response rates were estimated at up to 50% [44]. Simultaneously, the novel nitrogen mustard lomustine (CCNU) showed activity in patients with medulloblastoma [45,46]. Based on these observations, members of multiple collaborative groups (the Children’s Cancer Study Group and Radiation Therapy Oncology Group) proposed a trial to determine any potential benefit to adjuvant systemic chemotherapy following CSI.

The first prospective randomized trial to evaluate the role of adjuvant systemic chemotherapy enrolled patients age 2 to 16 years from 1975 to 1981 and compared CSI to CSI followed by chemotherapy with vincristine, lomustine, and prednisone [47]. Five-year event-free survival (EFS) was 55% and five-year overall survival (OS) was 65% and there was no survival difference between patients who did and did not receive chemotherapy; however, there was a benefit to chemotherapy in patients with advanced posterior fossa or metastatic disease.

A similar trial in Europe (SIOP) enrolled children under age 16 and compared CSI to CSI followed by vincristine and lomustine [48]. Overall, five-year survival was 53% and there was no survival benefit associated with chemotherapy. Despite the negative overall results, again there was a suggestion that certain patients, such as those with brainstem involvement, incomplete resections, or stage T3–4 disease, received benefit from chemotherapy.

Beginning in 1983 a series of clinically defined “high-risk” medulloblastoma patients treated at the University of Pennsylvania received CSI plus chemotherapy with vincristine, lomustine, and cisplatin [49]. Forty-two patients aged 3–21 years received the three-drug regimen, and actuarial analysis suggested a five-year disease-free survival of 85%—far superior to any previously reported. To validate these single institution results, 63 children aged 2–21 were subsequently enrolled to a multi-center study with the same regimen and had a five-year progression-free survival (PFS) of 85% [31]. Based on these results, this “Packer” regimen became the most widely used regimen in the United States.

The consistent observation of a benefit from chemotherapy in a subset of patients in the first two pre-cisplatin trials led to the hypothesis that the post-operative, pre-CSI interval provided an optimal window for chemotherapy. Presumed benefits included post-operative disruption in the local blood brain barrier allowing for increased delivery of chemotherapy to tumor cells and the increased ability to deliver myelosuppressive chemotherapy doses prior to CSI adversely affecting bone marrow function.

The SIOP II trial recruited patients younger than 16 years and compared postoperative CSI alone with post-operative chemotherapy (methotrexate, procarbazine, vincristine, and prednisolone) followed by CSI [50]. High-risk patients were also offered additional post-CSI chemotherapy with vincristine and lomustine, while low-risk patients were additionally randomized into standard vs low (36 vs. 25 Gy) CSI doses. Five-year overall survival was 59% and there was no significant improvement seen with chemotherapy. A second randomized trial, HIT’91, was conducted in Germany to compare post-operative pre-CSI “neoadjuvant” chemotherapy with post-CSI “maintenance” chemotherapy (the Packer regimen) and showed an OS benefit for patients with M0 or M1 disease treated with the Packer regimen [51].

The subsequent SIOP PNET-3 trial investigated if more intensive pre-CSI chemotherapy could improve outcomes for patients with nonmetastatic disease [32]. Patients from ages 3–16 received four cycles of vincristine and etoposide, with carboplatin or cyclophosphamide added on alternate cycles. Five-year overall survival was 70.7% and patients who received pre-CSI chemotherapy had significantly improved five-year EFS, 67.0% for chemotherapy plus CSI vs. 58.9% for CSI alone. This “Taylor” regimen has been used extensively since it was reported in 2003.

Since then, a series of prospective clinical trials in pediatric patients have further refined therapy for selected pediatric patients [52,53,54]. In contrast, similar studies have not taken place in adults. Treatment recommendations for adults derive from extrapolation from pediatric trials, retrospective analysis of adult cohorts, and a few prospective nonrandomized trials [16,33,55].

The first nonrandomized prospective trial published in adults with medulloblastoma stratified patients 18 or older into low and high-risk groups based on Chang’s scoring system and treated low risk patients (T1-3a and M0) with surgery and CSI and high risk patients (T3b-4 and M1-4) with surgery and CSI plus chemotherapy (either MOPP-like or cisplatin-based regimens) [56]. Initial results, published in 2007 and comprising 36 patients showed a five-year PFS and OS of 72% and 75%, respectively, with no difference in outcomes between low and high-risk patients [33]. However, an updated abstract reported in 2010 included 95 total patients and reported a 10-year OS of 65% in low-risk patients vs 45% in high-risk patients (*p* = 0.02) [57].

A second influential report resulted from a retrospective analysis of the pediatric HIT 2000 trial and analyzed outcomes in 70 adults (age 21+) who were treated with the Packer regimen [58]. Outcomes were good overall, with 4-year EFS of 68% and OS rates of 89%, and total resection was associated with a lower rate of progression. However, rates of chemotherapy toxicities such as peripheral neuropathy (74%) and hematologic toxicity (55%) were higher than those observed in children.

Following this experience, a pilot Phase 2 trial was undertaken by the German Neuro-Oncology Working Group (NOA) in adults with the goal of evaluating this regimen prospectively, particularly with regards to the toxicity profile [27]. Thirty patients older than 21 years of age were evaluated, and again neuropathic and hematologic toxicities were prevalent. Despite this, most (70%) patients were able to complete at least four cycles of treatment and three-year OS was 70%.

Retrospective meta-analyses provide another data source, with the largest reporting in 2016 on 907 individual “adults” (considered as 15 or older at time of diagnosis) from 227 studies from 1969 to 2013 [59]. Five-year OS was 50.9% and patients who received first-line chemotherapy (71%) had superior median OS (108 months, 95% Confidence Interval 68.6–148.4) than those who did not (29%) (57 months, 95% Confidence Interval 39.6–74.4). This conclusion was also supported by a review of the US National Cancer Data Base, which included 751 patients 18 and older who were treated between 2004 and 2012 [60]. Again, the majority (69.2%) received both chemotherapy and CSI and they had improved five-year OS (86.1% vs. 71.6%, *p* < 0.0001) when compared to those who received CSI alone (30.8%).

The landscape of clinical trials has changed significantly since the discovery of distinct molecular subgroups, and multiple targeted agents are currently being evaluated in clinical trials for both newly diagnosed and recurrent tumors. The first clinical trials utilizing genetic classification schemes are enrolling children (SJMB12-NCT01878617 and PNET5-NCT02066220) and adults (EORTC-1634-BTG-NCT04402073). The latter aims to enroll ~200 post-pubertal patients and randomize them to standard vs reduced-dose CSI and, for patients with SHH-activated tumors, to standard radio-chemotherapy with or without the SHH-pathway inhibitor, sonidegib.

The SHH-activated medulloblastomas have been a focus in research, largely due to long-standing interest in the pathway as an anti-cancer target. The first FDA approval for an SHH-pathway inhibitor was granted in 2012 for vismodegib for use in Basal Cell Carcinoma (sonidegib followed in 2015). The clinical experience in medulloblastoma has been limited to date but there is preliminary evidence of activity, with multiple responders in early-stage clinical trials [61,62]. SHH-pathway inhibitors are particularly appealing in adults due to the enrichment of SHH-activated disease as well as the absence of concern about one of their main adverse effects, toxicity to the developing skeleton.

This concept of age-dependent toxicity of novel drugs is important and raises a counterpoint to our consideration for age-independent trial inclusion. The developing nervous system is fragile in multiple poorly defined ways, and the risks of unwanted secondary effects are likely to be present with any novel targeted agents. We believe the best way to address this concern is through sound preclinical pharmacology work to address potential and observed toxicities and novel clinical development strategies encompassing the whole age spectrum, as will be discussed later with Larotrectinib.

There are two priorities for current and future clinical trials in medulloblastoma. The first is to identify patients with low-risk disease who are likely to be cured with current therapy and attempt to minimize acute and chronic toxicities from both radiation and chemotherapy by deescalating therapy. The second is to identify patients with aggressive disease who are at high-risk of poor outcomes and enroll them on clinical trials that utilize advanced molecular diagnostics and rationally targeted experimental therapeutics. We believe elevating molecular features of disease over arbitrary definitions such as pediatric vs adult will serve both goals, especially considering that an arbitrary cut-off of 18 or 21 years of age is inconsistent with the biology of the disease and responds more to organizational needs (i.e., pediatric versus adult practices).

One of the unfortunate realities of oncology is that despite a paucity of effective therapies for many disease types, only a small minority of eligible patients (<5%) are ever enrolled on the clinical trials that could help identify beneficial therapies [63,64]. As such, great efforts to design inclusive and accessible clinical trials for patients with primary CNS tumors are currently underway [65]. The medulloblastoma field, with its long history of international and collaborative trials, is well positioned to lead the way.

## 6. Models for Improvement

In arguing that molecular features of disease should be the primary factor in determining clinical trial eligibility, we can draw valuable lessons from three recent experiences. The observation that adolescent and young adults with acute lymphoblastic leukemia have improved outcomes when treated on “pediatric” chemotherapy regimens supports the hypothesis that selecting patients for clinical trials based on arbitrary age cutoffs is an inefficient and potentially harmful way to identify optimal regimens. The approval of the first tissue-agnostic cancer therapy, pembrolizumab in microsatellite-unstable tumors, suggests disease-associated molecular phenotypes may supersede other considerations for treatment. Finally, Larotrectinib, which was developed to target NTRK-fusion positive cancers and evaluated in a clinical trial program that included patients of all ages and tumor types, lends further credence to the supremacy of molecular features of disease as selection criteria for clinical trials.

### 6.1. Patient Age and Childhood Leukemia

Acute Lymphoblastic Leukemia (ALL) has a bi-modal age distribution, with ~60% of cases occurring in children younger than 5, a steady but low risk through adulthood, and an increasing incidence after age 50 [66].

ALL in children was a devastating diagnosis until the development of multi-agent chemotherapy regimens. Multi-institution collaborative groups came together to evaluate treatment regimens, and courses of chemotherapy now lead to five-year survival rates of 89% for patients diagnosed under age 20 [67]. However, the outlook remains poor for the ~40% of patients who are diagnosed after age 20, with five-year survival rates of 38% [66]. In older patients a desire to limit toxicities from therapies led to adults receiving lower doses of multiple medications (such as glucocorticoids, vincristine, and L-asparaginase), as well as shorter and less-intense CNS prophylaxis.

Retrospective analysis of multiple co-operative group trials led to the surprising observation that adolescents and young adults (AYA, ages 16–39) [68], who had variably been enrolled on either pediatric or adult clinical trials depending on trial-specific inclusion criteria, tended to have better outcomes when they were treated on pediatric regimens.

This observation led to a prospective trial to test the hypothesis that AYA would have improved clinical outcomes if they were treated with more-intensive pediatric regimens. CALBG 10403 enrolled over 300 patients from 2007 to 2012 and treated them with a pediatric standard-of-care regimen [69]. Results published in 2019 showed that treatment was well-tolerated and highly efficacious, with a three-year OS of 73% as compared to 55% for comparable patients treated on alternate studies.

Although the variable inclusion of patients of different ages on different trials did ultimately lead to the observation of improved outcomes in AYA patients treated with pediatric regimens, it did so by treating many patients on adult regimens with what we now know now to be sub-optimal care. Had the question been assessed prospectively it is likely the optimal regimen could have been determined with fewer patients needing to be treated, which could have prevented many poor outcomes associated with inferior treatment.

The lesson for the medulloblastoma field is that disease does not always vary significantly across the arbitrary age descriptors of childhood, adolescence, and adulthood. As such, clinical trials designed to include patients of all ages are likely the best way to quickly identify and implement optimal treatment regimens.

### 6.2. Pembrolizumab in MSI-Unstable/MMR-Deficient Tumors

Many cancers harbor defects in their DNA repair machinery such that they are unable to correctly copy short repeated sequences of DNA known as microsatellites [70]. Tumors bearing this phenotype are referred to as microsatellite instability high (MSI-H) or mismatch repair deficient (dMMR). This phenomenon is most observed in colorectal, gastric, and endometrial cancers but it can be observed in tumors arising from any organ.

Data from multiple clinical trials with pembrolizumab, an anti-PD-1 antibody, led to the observation that many patients with MSI-H/dMMR tumors had excellent responses to treatment. Of 149 patients with nine different cancers treated with pembrolizumab, the overall response rate was 39.6% [71], an excellent rate for second line treatment of metastatic solid tumors. Based on this data, in 2017 the FDA took the novel step of approving pembrolizumab for all patients with metastatic solid-tumors with the MSI-H/dMMR molecular phenotype regardless of tissue origin [72]. This indication was also extended to pediatric patients based on the “scientific rationale and the establishment of a reasonably safe dose in other pediatric clinical trials”.

The analogy to medulloblastoma patients can be seen in considering subgroups of disease as specific molecular phenotypes. As our ability to quantify molecular drivers of disease improves, the hope is that we will be able to identify specific molecular phenotypes that will predict responsiveness to targeted therapies.

### 6.3. Larotrectinib in TRK-Fusion Positive Tumors

The neurotrophic receptor kinase genes NTRK1, 2, and 3 encode a family of receptor tyrosine kinases that function during the development of the nervous system. All three genes are susceptible to chromosomal fusion events that result in the intracellular kinase domain of the TRK receptors being linked to various partners and lead to uncontrolled kinase activity. These unregulated growth signals lead to oncogene addiction, drive malignant behavior in a variety of tissue types and patient populations, and are seen in up to ~1% of all solid tumors [73].

Larotrectinib is a highly selective small molecule inhibitor of all three TRK proteins [74]. It was evaluated in a novel program that encompassed patients of all ages and tumor types. The primary criteria for inclusion on trial was tumor positivity for a TRK-fusion event, regardless of what organ it originated in. A total of 55 patients were treated across three protocols, including both adults and children and encompassing 17 unique TRK-positive fusion types.

Results were impressive, with an overall response rate of 75%, of which 71% were ongoing after one year of treatment. Larotrectinib received accelerated approval from the FDA in November of 2018 for adult and pediatric patients with solid tumors that have a NTRK gene fusion.

The lesson from the story of TRK-fusions and Larotrectinib is that the most salient features of tumors are the molecular alterations that drive their aggressive behavior. By designing a clinical trial program that prioritized these molecular features over age or tissue of origin, the investigators were able to identify, validate, and receive approval for an effective therapy for patients within an accelerated timeline, proceeding from receiving orphan drug status to FDA approval in just three years (2015–2018).

## 7. Challenges

The challenges to enacting our proposal are numerous and significant. The single largest will be overcoming the binary division of care whereby adult and pediatric patients are largely treated by different teams of providers or even at different hospitals. Furthermore, unequal access to specialty care remains a significant problem for patients in the USA. Designing and implementing trials enrolling patients across the age spectrum will require considerable investment from multiple stakeholders, including academic medical centers, pharmaceutical companies, and collaborative groups. Trial design must incorporate clear guidelines for stratification based on molecular markers, and potentially by age if the profile of side effects is expected to be different, as well as strict rules for monitoring of toxicity and dose modifications. The international brain tumor community, most notably the International Society for Pediatric Oncology in Europe and the Children’s Oncology Group in the USA, has demonstrated a willingness to implement rational clinical trials based on molecular subgroup-guided stratification strategies (NCT01878617, NCT02066220), and it is our hope that continued, dedicated effort will continue.

## 8. Conclusions

In summary, although the outcomes of patients with medulloblastoma have improved markedly over the last few decades, these have come primarily from improvements in surgical management and non-specific chemotherapy and radiation therapies. As our molecular understanding of the disease has improved over the past decade, we can now identify specific subsets of patients with divergent clinical prognoses. Despite these diagnostic advances, the clinical trials to date have largely evaluated chemotherapy and radiation modifications for pediatric patients. Lessons from clinical trials and targeted therapies in other fields have consistently shown the benefits to considering molecular drivers of disease as primary features for clinical trial eligibility, regardless of age. It is our hope that moving forward, prospective clinical trials enroll medulloblastoma patients of all ages based on molecular features of disease and identify safe and effective treatments.

## Figures and Tables

**Figure 1 cancers-13-06313-f001:**
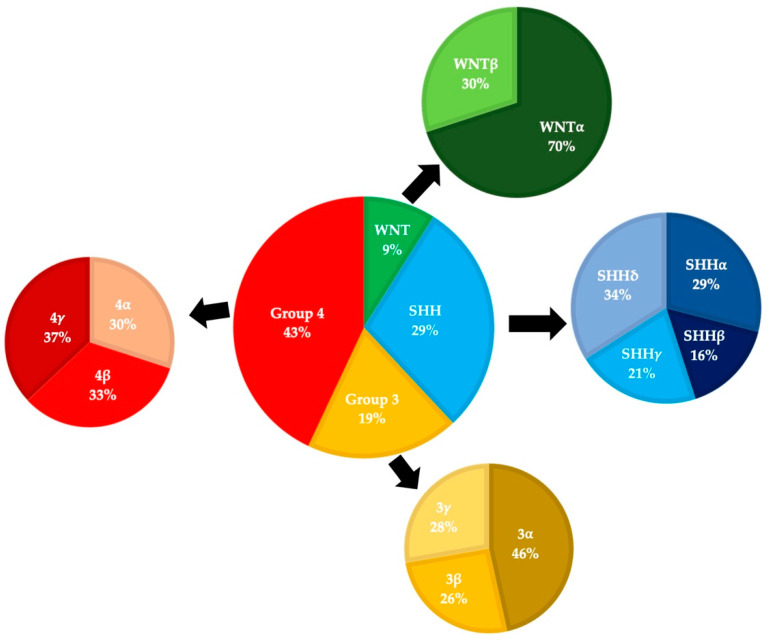
Medulloblastoma molecular subgroups and subtypes pie charts are based on data from Cavalli (2017). This series included 763 patients, of which 101 were adults [9].

**Figure 2 cancers-13-06313-f002:**
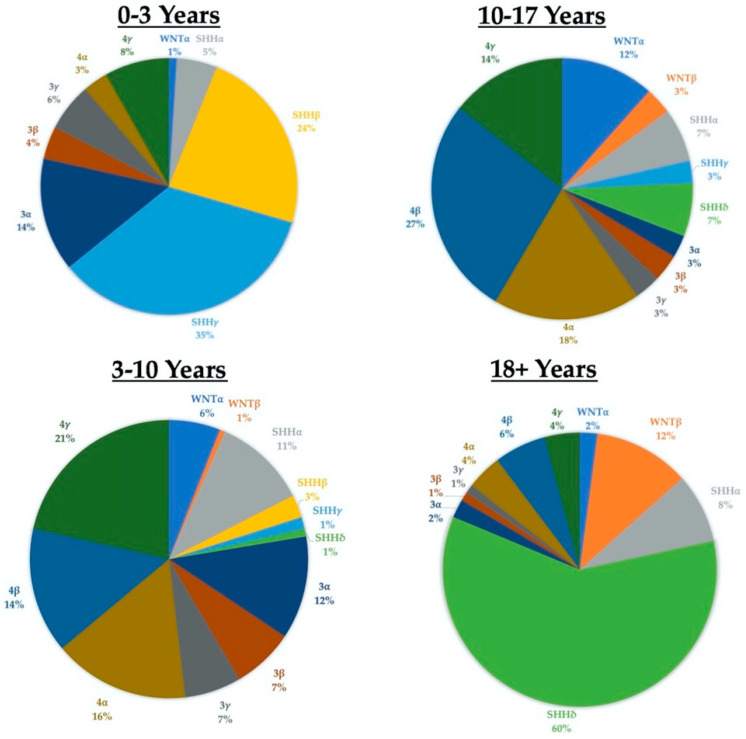
Subgroups and subtypes across the age spectrum.

**Table 1 cancers-13-06313-t001:** Clinical Trials Available to Older Patients (30 years+): less than 20% of all trials recruiting patients with medulloblastoma.

Trial #	Year Opened	Phase	Patient # (Planned)	Ages (Years)	Eligibility	Treatment
NCT04315064	2020	1	5	1–80	Recurrent/Progressive MB	Panobinostat (intrathecal)
NCT04315064	2019	1/2	60	3 and up	Recurrent SHH MB	CX-4945
NCT01878617	2013	2	625	3–39	Any MB age 3–22, SHH MB 22–39	Chemo, RT, Vismodegib
NCT01857453	2013	2	97	18–70	Standard risk adult MB	Chemo + reduced dose RT
NCT02962167	2017	1	46	12–39	Recurrent MB or ATRT	Modified Measles virus (MV-NIS)
NCT04661384	2021	1	30	18+	Recurrent Leptomeningeal MB, GB, or Ependymoma	IL-13Ralpha-2 CAR-T Cells
NCT03893487	2019	1	30	3–39	DIPG, Recurrent MB of HGG	Fimepinostat
NCT03434262	2018	1	108	1–39	Recurrent MB + other CNS Tumors	Ribociclib + gemcitabine, trametinib, or sonidegib
NCT03173950	2017	2	180	18+	Recurrent MB + others	Nivolumab
NCT03734913	2019	1	65	18–75	Advanced MB + others	ZSP1602
NCT04541082	2020	1	102	18+	Recurrent MB + others	ONC206
NCT01505569	2011	2	20	Up to 70	Recurrent MB + others	Chemo + Autologous HSCT
NCT02905110	2016	1	10	1–80	Recurrent MB + other PF Tumors	Etoposide and Methotrexate (intrathecal)

Abbreviations: Trial #: National Clinic Trial number from ClinicalTrials.gov, Patient #: Number of planned patients to be enrolled, MB: Medulloblastoma, RT: Radiation Therapy, MV-NIS: Measles Virus Sodium Iodide Symporter, GB: Gliobastoma, CAR-T: Chimeric antigen Receptor T-cell, ATRT: Atypical Teratoid Rhabdoid Tumor, DIPG: Diffuse Intrinsic Pontine Glioma, HGG: High-Grade Glioma, HSCT: Hematopoietic Stem Cell Transplant, PF: Posterior Fossa.

## Data Availability

This study does not report any previously unpublished data.

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
