# Peer review of "Pediatric versus Adult Medulloblastoma: Towards a Definition That Goes beyond Age"

_cancers, 2021, doi:10.3390/cancers13246313_

Round 1

Reviewer 1 Report

To authors

This is a very interesting manuscript focused on pediatric and adult medulloblastoma. I recommend this manuscript for publication after some below minor revisions

1) This is a review; thus, unstructured abstract is precise. Please revise abstract

2) Added diagnostic imaging part for medulloblastoma and consider to add below up-to-date references. Noted that also share information about the discrimination between medulloblastoma and other primary CNS neoplasms such as ependymoma, pilocytic astrocytoma, DIPG,...

- The Role of Apparent Diffusion Coefficient in the Differentiation between Cerebellar Medulloblastoma and Brainstem Glioma. Neurol Int. 2020 Oct 29;12(3):34-40. doi: 10.3390/neurolint12030009. PMID: 33137983; PMCID: PMC7768368.

3) Authors only discuss about molecular subtypes. I suggest one more part about conventional subtypes of medulloblastoma including classic, desmoplastic, large cell/anaplastic, nodularity. 

4) The citation format is completely wrong as MDPI format. Please check author guideline and revise

5) The reference forma is partially wrong as MDPI format. Please check author guideline and revise.

Sincerely

Reviewer 2 Report

Medulloblastoma has been classified into subtypes based on histopathological features. But, recent revolution in genomic technologies has led to the molecular classification of MB (PMID:33159234, well described in this mini-review).Historically, arbitrary age cutoffs have been used in clinical studies, whereas trials based on molecular profiling of disease and including older adults have been scarce.

In the  present article entitled “Pediatric versus Adult medulloblastoma: towards a definition that goes beyond age” the authors argue that molecular features of the disease should be given consideration over age and they provide examples of various clinical trials to substantiate their argument.

Overall the manuscript is straightforward, well written, concise, and clear within the scope of  MDPI-Cancers.

I am okay with the manuscript to be published as it is.

Reviewer 3 Report

The authors present an interesting article on pediatric vs. adult medulloblastoma.

-While being a mix of a systematic review and an overview of currently ongoing trials the authors fail to adhere to prisma guidelines.

-What is the point of table 2? The authors present a list of ongoing studies without addressing them in detail. They present 13 studies including adults. Why? What is the clinical message? Should these studies open to paediatric patients?

-The authors want to move "beyond age as a stratifyer" however, the data they provide displays differences in chemo side effects between adults and paediatric patients. These findings rather support such strategy of age-specific therapy (while obviously molecular markers should be considered for targeted therapy).

-Further, the models for improvement focus on three observations not closely related to medulloblastoma research. The authors should address successes in molecular therapy addressing markers/mutations commonly observed in medulloblastoma

Reviewer 4 Report

Review

The review manuscript by Joseph R Wooley and Marta Penas-Prado discusses medulloblastoma with a focus on the relevance of age versus molecular profile in considering patient enrolment in clinical trials. The subject should be of interest to the readership of Cancers.

Major critics:

-The major goal and the rational for this goal are not clearly stated in the abstract, in my opinion. For example, a clear description of the molecular characterization of MB in the past 10 years, i.e., describe the 4 (or more) well-defined major subtypes, should be described as part of the rational. Also, the "Result" section of the abstract does not describe results per se.

-While the reviewer totally agrees with the authors that the potential benefit of targeted therapy are likely to be age-independent, one could argue that one major age-dependent limitation associated with any given targeted therapy are the potential secondary, unwanted effects associated with the therapy. Indeed, these secondary effects may have a more detrimental impact on the developing brains of children than on the ones of adults. The reviewer feels that these considerations, which could counter-argue for the need for an age-dependent stratification of patients in clinical trials, not so much because of the potential benefits but rather the risks, should, at least, be mentioned and discussed in this review.

-Given the main focus of the review , i.e., to discuss the need to stratify MB patients based on age in clinical trials or not, the lengthy historical review of clinical trials in MB described in Section #4 is not always, on point. I would suggest shortening it and mentioning the observations that are relevant to the main focus of this review: age versus molecular profile.

-It would have been good to describe the historical rationales for treating adult and pediatric patients differently and then argue why some of these rationales are not always applicable or could be misleading.

Minor suggestions:

-"Medulloblastoma" is uppercased, sometimes. Sometimes not. No need to uppercase, I think, but be consistent throughout text.

-"Pediatric versus Adult medulloblastoma: towards a definition  that goes beyond age."

No need to uppercase "Adult".

-"reports we found influential." Seems a bit subjective. Any more objective criteria to mention: size, impact on community, discovery, game changer...

-"We performed a clinicaltrials.gov search for ongoing clinical trials enrolling adults with medulloblastoma."

This sentence suggests that trials enrolling pediatric patients were ignored. Yet the focus is pediatric vs adult. A bit confusing.

-"sizable minority": This oxymoron could describe any data point below 50%. Use more a more precise estimate.

-There are a few issues with the reference formatting: {Robinson, 2020 #990}{Cavalli, 2017 71 #991}

-Table 1: a graph representation of the data would have been more useful and visually appealing.

-"Although rare overall, these numbers are broadly on the scale with other rare cancers (such as malignant ovarian teratomas, ~500/cases/year in US) (Botta et al., 2020) and well-known genetic disorders such as Spinal Muscular Atrophy (~400 births per year (Keinath, Prior, & Prior, 90 2021)) and Cystic Fibrosis (~1,000 new diagnoses per year (Elborn, 2016))."

This sentence does not add much, could be deleted.

-Many spelling/punctuation mistakes could have been caught by a simple review (both manual and "spelling and grammar" software check.

-"and it is out hope that continued, dedicated effort will continue."

-Some "periods" . are missing.

These observations suggest that proof-reading was not performed.

Round 2

Reviewer 3 Report

The authors have extensively reviewed the manuscript. Hence I do not have objections against publication.